# Yohimbine Inhibits PDGF-Induced Vascular Smooth Muscle Cell Proliferation and Migration via FOXO3a Factor

**DOI:** 10.3390/ijms25136899

**Published:** 2024-06-24

**Authors:** Leejin Lim, Hyeonhwa Kim, Jihye Jeong, Sung Hee Han, Young-Bob Yu, Heesang Song

**Affiliations:** 1Advanced Cancer Controlling Research Center, Chosun University, Gwangju 61452, Republic of Korea; 2Department of Biomedical Sciences, Chosun University Graduate School, Gwangju 61452, Republic of Korea; 3Institute of Human Behavior & Genetics, Biomedical Research Center, Korea University, Seoul 02841, Republic of Korea; 4Department of Paramedicine, Nambu University, Gwangju 62271, Republic of Korea; 5Department of Biochemistry and Molecular Biology, Chosun University School of Medicine, Gwangju 61452, Republic of Korea

**Keywords:** yohimbine, proliferation, migration, vascular smooth muscle cells, FOXO3a, mTOR

## Abstract

Yohimbine (YHB) has been reported to possess anti-inflammatory, anticancer, and cardiac function-enhancing properties. Additionally, it has been reported to inhibit the proliferation, migration, and neointimal formation of vascular smooth muscle cells (VSMCs) induced by platelet-derived growth factor (PDGF) stimulation by suppressing the phospholipase C-gamma 1 pathway. However, the transcriptional regulatory mechanism of YHB controlling the behavior of VSMCs is not fully understood. In this study, YHB downregulated the expression of cell cycle regulatory proteins, such as proliferating cell nuclear antigen (PCNA), cyclin D1, cyclin-dependent kinase 4 (CDK4), and cyclin E, by modulating the transcription factor FOXO3a in VSMCs induced by PDGF. Furthermore, YHB decreased p-38 and mTOR phosphorylation in a dose-dependent manner. Notably, YHB significantly reduced the phosphorylation at Y397 and Y925 sites of focal adhesion kinase (FAK), and this effect was greater at the Y925 site than Y397. In addition, the expression of paxillin, a FAK-associated protein known to bind to the Y925 site of FAK, was significantly reduced by YHB treatment in a dose-dependent manner. A pronounced reduction in the migration and proliferation of VSMCs was observed following co-treatment of YHB with mTOR or p38 inhibitors. In conclusion, this study shows that YHB inhibits the PDGF-induced proliferation and migration of VSMCs by regulating the transcription factor FOXO3a and the mTOR/p38/FAK signaling pathway. Therefore, YHB may be a potential therapeutic candidate for preventing and treating cardiovascular diseases such as atherosclerosis and vascular restenosis.

## 1. Introduction

The excessive proliferation and migration of vascular smooth muscle cells (VSMCs) are major factors contributing to the progression and development of cardiovascular diseases such as restenosis and atherosclerosis [1]. Abnormalities in VSMCs occur in response to various stimuli, such as platelet-derived growth factor (PDGF), oxidative damage, and cytokines from vascular endothelial cells, activated platelets, or immune cells [2]. PDGF leads to the production and secretion of the extracellular matrix (ECM), resulting in vascular intimal hyperplasia [3]. Simultaneously, it induces cell proliferation that forms new vascular endothelium on the inner walls of blood vessels [4,5]. This process triggers arterial narrowing and disturbances in blood flow, thereby mediating the initial response to atherosclerosis.

The interaction between PDGF and its receptor regulates the migration and proliferation of VSMCs by mediating various signaling pathways, including the mitogen-activated protein kinase (MAPK) family, which includes extracellular-regulated protein kinase (ERK), c-Jun N-terminal protein kinase (JNK), p38 MAPK, or Akt [2]. The activation of p38 MAPK is involved in the proliferation of VSMCs in response to thrombin and balloon injury [6,7,8].

Yohimbine (YHB) is a natural indole alkaloid extracted from various biological sources, including *Pausimystalia yohimbe* bark and *Rauwolfia* root [9]. The effects of YHB hydrochloride as a selective and potential α2-adrenergic receptor antagonist make it useful not only for treating erectile dysfunction but also for weight loss, muscle toning exercises, and enhancing the effects of antidepressants [10,11]. Research suggests that YHB has effects in improving lipopolysaccharide (LPS)-induced gastrointestinal disorders and acts as an analgesic for α-receptor agonists, including xylazine or detomidine, by enhancing neural leakage via the release of the neurotransmitter norepinephrine in the nervous system [12,13]. It has also been reported to inhibit melanin synthesis in the skin by downregulating microphthalmia-associated transcription factor (MITF) and tyrosinase expression through the regulation of the Wnt/β-catenin and p38/MAPK signaling pathways [14]. Additionally, YHB enhanced the therapeutic effects of berberine (an antibiotic) against bacteremia induced by LPS, by upregulating interleukin (IL)-10. It achieves this by inhibiting various signaling pathways such as JNK, ERK, and NF-kB, among others. YHB has also been shown to prevent LPS-induced sepsis associated with kidney damage by suppressing inflammatory cytokines [15,16]. Moreover, it is known to enhance anti-inflammatory effects in a rat model of arthritis by modulating post-inflammatory cytokine inhibition and antioxidant status [17]. Furthermore, YHB displays anticancer effects by inducing apoptosis in breast cancer and pancreatic cancer cells, thereby inhibiting cell proliferation [18,19]. Furthermore, YHB increases the release of norepinephrine within the heart by blocking presynaptic α2A-adrenergic receptors at the synapse, partially alleviating the cardiac dysfunction induced by LPS. This action inhibits cardiomyocyte apoptosis, which suppresses the expression of inducible nitric oxide synthase (iNOS) and tumor necrosis factor (TNF)-α in the heart, enhancing cardiac function [20]. However, there are also reports suggesting that YHB induces myocardial damage, with controversial results in cardiovascular diseases [21]. Furthermore, α2-adrenergic receptors can regulate vasoconstriction in VSMCs and potentially play a role in macrophage function and the progression of arteriosclerosis. Research is being conducted to develop drugs that can treat or inhibit cardiovascular diseases. However, given the known side effects of synthetic therapeutic agents, there is a growing focus on the development of natural-ingredient pharmaceuticals that are safer, have fewer side effects, and are more effective. Medical plant research has evolved beyond the simple isolation of chemical compounds and drug efficacy testing. Studies have revealed the fundamental mechanisms underlying the remarkable therapeutic effects of medicinal plants. This understanding has significantly contributed to the increased use of medicinal plants or derivatives from natural sources for the development of pharmaceuticals [22].

Based on this, YHB has been reported to inhibit VSMC proliferation, migration, and neointimal formation stimulated by PDGF by blocking the phospholipase C-gamma 1 (PLC-γ1) pathway, which is involved in the regulation of cell signaling [23]. However, the transcription factors and their related signaling pathways involved in the inhibitory effects of YHB on the proliferation and migration of VSMCs induced by PDGF-BB have not yet been reported. In this study, we demonstrated, for the first time, the inhibitory effects of YHB on the proliferation and migration of PDGF-BB-stimulated VSMCs by regulating the transcription factor Forkhead box O3a (FOXO3a) and the mTOR/p38/FAK signaling pathway.

## 2. Results

### 2.1. YHB Inhibits PDGF-BB-Induced VSMC Proliferation and Migration

YHB (C_21_H_26_N_2_O_3_; MW: 390.90; purity: HPLC ≥ 98%) was obtained from Sigma-Aldrich (St. Louis, MO, USA), and its chemical structure is shown in Figure 1A. To determine the experimental concentration of YHB, its cytotoxicity was evaluated using the 3-(4,5-dimethylthiazol-2-yl)-2,5-diphenyltetrazolium bromide (MTT) assay. As observed in Figure 1B, treatment with 25–100 μM YHB did not affect the viability of VSMCs, whereas concentrations of 200–400 μM exhibited cytotoxicity on these cells. Therefore, we evaluated the beneficial effects of YHB on the abnormal behavior of VSMCs under a 100 μM concentration. As shown in Figure 1C, at a concentration of 50 μM YHB, an inhibitory effect on the proliferation of VSMCs was observed. Although treatment with 100 µM was effective, no significant difference was observed with 50 µM. Therefore, we chose 10 µM and 50 µM for subsequent experiments. We evaluated the effect of YHB on the migration using a 2D wound healing assay. We observed a significant increase in the migration of VSMCs treated with PDGF-BB compared to the control group, which was significantly inhibited by YHB in a dose-dependent manner (Figure 1D). These results suggest that YHB inhibits the PDGF-BB-induced migration and proliferation of VSMCs.

### 2.2. YHB Suppresses PDGF-BB-Induced Expression of Cell Cycle Regulatory Proteins in VSMCs

To investigate the inhibitory mechanism of YHB on the proliferation of VSMCs, cell cycle progression was examined using fluorescence-activated cell sorting (FACS), and the expression of cell cycle-regulating proteins was assessed by Western blotting and immunofluorescence. FACS analyses showed a significant dose-dependent increase in the G0/G1 phase cell population and a decrease in the S phase cell population in YHB-administered VSMCs stimulated with PDGF-BB (Figure 2A). Furthermore, the protein expression of PCNA, CDKs, and cyclins, well-known markers for proteins essential for DNA synthesis and cell proliferation, was evaluated. As shown in Figure 2B, the increased levels of PCNA, cyclin D1, CDK4, and cyclin E following PDGF-BB treatment were significantly inhibited by YHB in a dose-dependent manner. Similar immunofluorescence results were observed, as shown in Figure 2C. These results suggest that YHB inhibits the proliferation of VSMCs by arresting the cells in the G0/G1 phase and suppressing the expression of cell cycle regulatory proteins.

### 2.3. YHB Inhibits PDGF-BB-Induced VSMC Proliferation via Regulation of FOXO3a Factor

Several transcription factors, including FOXO3a, regulate target proteins to promote VSMC proliferation and migration [24,25]. We hypothesized that FOXO3a is a potential target of YHB in PDGF-BB-treated VSMCs. As expected, increased expression levels of FOXO3a in PDGF-BB-treated VSMCs were significantly inhibited by YHB treatment in a dose-dependent manner. We found that the expression levels of E2F-1, another transcription factor known to regulate the cell cycle, were significantly decreased with YHB treatment (Figure 3A). YHB treatment further decreased the expression levels of PCNA and cyclin D1 in FOXO3a-silenced VSMCs (Figure 3B). However, no significant changes in the expression of cell cycle regulatory proteins were observed after YHB treatment when E2F-1 was silenced. These results suggest that the decrease in the expression levels of cell cycle regulatory proteins in VSMCs following YHB treatment occurs via FOXO3a, which acts as a regulatory transcription factor.

### 2.4. YHB Modulates FAK-Related and mTOR/p38 Signaling Pathway in PDGF-BB-Stimulated VSMCs

Many studies have reported that FAK and related protein complexes participate in VSMC proliferation and migration [26,27]. We observed that the phosphorylation levels at the Y397 and Y925 sites of FAK were significantly increased after treatment with PDGF-BB and were significantly inhibited by YHB. In particular, a successful withdrawal effect was observed at the Y925 site due to YHB. Additionally, the levels of FAK-related proteins that bind to the Y925 site of FAK, such as paxillin, decreased upon YHB treatment in a dose-dependent manner (Figure 4A). Other FAK-related proteins, such as vinculin, talin, and p-Src, did not show any significant changes following YHB treatment. Furthermore, the significantly increased levels of p38 and mTOR phosphorylation observed after PDGF-BB treatment were reduced by YHB treatment. However, YHB treatment did not decrease the phosphorylation of ERK and Akt following PDGF-BB treatment (Figure 4B). These results suggest that the inhibition of cell migration and proliferation of VSMCs by YHB treatment occurs through the FAK-related and p38/mTOR pathways.

### 2.5. YHB Inhibits PDGF-BB-Induced VSMCs Proliferation and Migration Synergistically with mTOR/p38 Signaling Inhibition

To determine the effect of cotreatment with YHB, mTOR, and p38 inhibitors on the migration of VSMCs, we performed a 3D Boyden chamber assay. As shown in Figure 5A, the increased migration of VSMCs induced by PDGF-BB was markedly reduced in VSMCs treated with YHB. Moreover, we observed that the decreased levels were significantly higher when mTOR and p38 inhibitors were co-treated with YHB than when YHB was used alone. In particular, the inhibitory effect of the p38 inhibitor was greater than that of the mTOR inhibitor. Furthermore, to confirm whether the inhibitory effect of YHB on the expression of cell cycle regulatory proteins induced by PDGF-BB occurred through the mTOR and p38 signaling pathways, the expression of Forkhead box O3a (FOXO3a), cyclin D1, and PCNA was assessed. As shown in Figure 5B, the increased expression levels of FOXO3a, cyclin D1, and PCNA induced by PDGF-BB were significantly reduced by the mTOR inhibitor only and by co-treatment with YHB and the mTOR inhibitor. The degree of inhibition from the co-treatment was significantly higher than when the mTOR inhibitor was used alone. When the p38 inhibitor was administered alone, the expression levels of FOXO3a and cyclin D1 were not significantly decreased, whereas they were significantly decreased by co-treatment with YHB and the p38 inhibitor. The expression of PCNA remained unchanged, even in VSMCs co-treated with YHB and a p38 inhibitor. These results suggest that YHB inhibits PDGF-BB-induced VSMC migration through the p38 and mTOR signaling pathways and proliferation through the mTOR signaling pathway.

## 3. Discussion

The abnormal proliferation and migration of VSMCs leading to intimal hyperplasia are known to cause cardiovascular diseases such as atherosclerosis and restenosis after angiography [28]. Among the several stimuli responsible for this phenomenon, PDGF-BB plays a crucial role in increasing the proliferation and migration of VSMCs, leading to the development of many cardiovascular diseases, including atherosclerosis and restenosis [29].

YHB has potential therapeutic applications in various diseases owing to its anti-inflammatory [16,17], anticancer [18,19], and cardioprotective effects [20]. Even though YHB is also known to inhibit the proliferation, migration, and neointima formation of VSMCs induced by PDGF-BB by inhibiting the PLC-γ1 pathway [23], the related signaling mechanism and transcription factor remain elusive. In this study, we proposed that YHB may serve as a potential therapeutic agent by inhibiting the proliferation and migration of VSMCs induced by PDGF-BB, thus preventing arteriosclerosis and restenosis due to the following: (i) YHB inhibits the proliferation of VSMCs by downregulating cell cycle regulatory proteins; (ii) YHB regulates the proliferation and migration of VSMCs via targeted signals such as mTOR, p38, and FAK; (iii) YHB inhibits the expression of cell cycle regulatory proteins through the regulation of FOXO3a. The results are shown in a schematic diagram in Figure 6. Specifically, YHB inhibits the phosphorylation of p38 MAPK and mTOR, among the various signaling molecules activated by PDGF-BB, resulting in the suppression of PCNA, cyclin D1, CDK4, and cyclin E expression. These cell cycle regulatory proteins are overexpressed in PDGF-BB-induced VSMCs and promote cell proliferation [30,31]. Additionally, PDGF-BB activates many downstream signaling molecules related to cell growth, survival, and proliferation, such as the PI3K, Akt, PLC-γ1, and ERK1/2 pathways [32]. Rapamycin, an mTOR pathway inhibitor, inhibits human and rat aortic SMC proliferation and migration by suppressing cell cycle-dependent kinases and delaying the phosphorylation of retinoblastoma proteins in vitro [33,34]. Furthermore, treatment with rapamycin significantly reduced the progression of transplant vasculopathy in patients undergoing heart surgery [35]. Rapamycin also inhibited intimal hyperplasia in animal models and human clinical trials by inducing the expression of p21 and p27, which are cyclin-dependent kinase inhibitors [36].

The FOXO family of transcription factors plays a crucial role in regulating various cellular processes, including cell survival, apoptosis, cell cycle arrest, proliferation, and migration [37]. Several studies have suggested that phosphorylation of FOXO3a plays a crucial role in mediating the migration of VSMCs by inhibiting TIMP-2 and increasing the expression of MMP-2 [38]. In addition, FOXO3a significantly increases MMP-13 expression levels and promotes VSMC migration [24]. The mTOR signaling pathway controls cell proliferation by regulating FOXO3a [37]; however, the relationship between mTOR and FOXO3a in VSMCs is not well established. In this study, we confirmed that PDGF-BB induced the phosphorylation of Akt, ERK1/2, p38, JNK, and mTOR. However, YHB significantly reduced the phosphorylation levels of p38 and mTOR but had no effect on other activated signaling molecules (Figure 4B). Furthermore, previous research reported that YHB inhibits the proliferation, migration, and neointimal formation of murine VSMCs stimulated by PDGF by blocking the PLC-γ1 pathway without affecting ERK1/2, Akt, and p38 phosphorylation [23], which is not consistent with our results. We observed a significant inhibition of p38 phosphorylation in PDGF-BB-treated VSMCs treated with YHB. The discrepancies between the previous studies and the present study are attributable to the different cell types and experimental conditions used. Furthermore, our study demonstrated for the first time that FOXO3a and E2F1, which are overexpressed in VSMCs induced by PDGF-BB, were significantly and dose-dependently inhibited by YHB. In particular, a synergistic effect was observed, in which the knockdown of FOXO3a enhanced the inhibitory effect of YHB on cell proliferation (Figure 3B). Moreover, these transcription factors act as downstream targets of mTOR [39]. Previous research has suggested that after simvastatin treatment, mTOR/FOXO3a activation promotes the proliferation and migration of cardiac microvascular endothelial cells [40]. mTOR is a major regulator of cellular metabolism that controls cell growth and proliferation and plays a crucial role in protein synthesis [41]. Indeed, we demonstrated that the increased expression levels of FOXO3a, cyclin D1, and PCNA induced by PDGF-BB were significantly reduced by co-treatment with YHB and an mTOR inhibitor (Figure 5B).

Interactions between cells and the ECM are necessary for cell migration [42]. FAK is a non-receptor cytosolic tyrosine kinase that regulates various physiological processes, including cell migration, proliferation, and survival. The activation of FAK is known to occur through integrin signaling, which is activated by the extracellular matrix or growth factor signals [43]. Previous studies have reported FAK overexpression in highly metastatic cancer tissues, such as the liver, breast, and thyroid [44,45]. In addition, FAK has been identified as a critical enzyme that regulates the migration and proliferation of VSMCs in important pathological processes related to cardiovascular diseases such as atherosclerosis and vascular restenosis [46]. In the current study, we observed a significant inhibition of FAK phosphorylation at the Y397 and Y925 sites and the expression levels of paxillin, a FAK-related protein, in YHB-treated VSMCs (Figure 4A). Additionally, a synergistic effect was observed in which treatment with the p38 inhibitor enhanced the inhibitory effect of YHB on cell migration (Figure 5A). We propose that reduced p38 MAPK signaling due to YHB treatment inhibits FAK phosphorylation, leading to the suppression of VSMC migration. Furthermore, this study revealed that YHB inhibits the migration and proliferation of VSMCs by inhibiting FAK activity via p38 and reducing FOXO3a through mTOR signaling.

Therefore, we propose that YHB is a promising candidate for the prevention and treatment of cardiovascular diseases such as atherosclerosis and vascular restenosis. Further experiments on animal models are warranted to determine the therapeutic effect of YHB in cases where atherosclerosis is induced. Moreover, future animal model studies should also focus on cross-validating the signaling system presented in this study. In addition, we suggested that transcription factors play a significant role in the regulation of VSMCs during the progression of atherosclerosis. Establishing a new field of diagnosis and treatment for atherosclerosis based on the association between transcription factors and VSMCs may lead to the development of novel therapeutic targets and drugs for atherosclerosis.

## 4. Materials and Methods

### 4.1. Isolation and Culture of VSMCs

Rat aortic VSMCs were isolated from 6-week-old Sprague-Dawley rats [28]. Briefly, the thoracic aorta was dissected, and loose connective tissue around the blood vessels was removed. The aorta was transferred to a tube containing a mixture of collagenase type I (1 mg/mL; Sigma-Aldrich) and elastase (0.5 mg/mL; Worthington Biochemical Co., Worthington, NJ, USA) and cultured at 37 °C for 30 min. Subsequently, the aorta was placed in a 100 mm cell culture dish, and the outer layer was peeled off under a microscope. The obtained medial layer was finely chopped and placed in a tube containing 5 mL of enzyme dissociation mixture (including collagenase and elastase) and incubated at 37 °C for 2 h. The pellet obtained after centrifugation at 1600× *g* for 5 min was resuspended in DMEM containing 10% fetal bovine serum (FBS) (Welgene, Gyeongsan, Republic of Korea). This step was repeated until the tissues were completely dispersed. Cells were cultured in a 37 °C incubator with 5% CO_2_, and cells from passages 5 to 13 were used in this study.

### 4.2. Cell Viability Assay

Cell viability and cytotoxicity were measured using the MTT assay. VSMCs were seeded in a 96-well plate at a density of 1 × 10^4^ cells/well and cultured at 37 °C with 5% CO_2_ for 24 h. After culturing, samples were prepared at various concentrations and treated for 24 h. The control group was cultured under the same conditions with an equivalent amount of DMSO. MTT solution at a final concentration of 0.5 mg/mL was added, and the cells were incubated for 2 h at 37 °C. Absorbance was measured at 490 nm using a microplate reader (spectrophotometer). The experiment was performed in triplicate.

### 4.3. Cell Proliferation Assay

The proliferation of VSMCs was assessed using the 5-bromo-2′-deoxyuridine (BrdU) incorporation assay. The cells were cultured in a 96-well microtiter plate at a density of 2 × 10^3^ cells/well for 12 h. The cells were starved for 6 h in DMEM without FBS. The cells were treated with YHB or PDGF-BB for 24 h. BrdU-labeling solution (10 µM) was added to the cells and incubated for 12 h to label newly synthesized DNA. The amount of newly synthesized DNA was quantified. DNA quantification involved denaturing the DNA, followed by a 90 min incubation at room temperature (RT) with a peroxidase-labeled anti-BrdU monoclonal antibody. Subsequently, the amount of BrdU-antibody complex was detected using a Victor 3 luminometer (PerkinElmer, Waltham, MA, USA).

### 4.4. Cell Cycle Analysis

Cultured cells were maintained in a nutrient-deprived state (0.1% FBS) for 24 h, followed by treatment with YHB. After 2 h, cells were treated with PDGF and cultured for an additional 24 h. The cells were then rinsed, collected using 0.5× Trypsin-EDTA, and dissolved in 300 µL PBS. Next, 70% ethanol was added dropwise to the cells, which were incubated at 4 °C in a chamber for 1 h. The fixed cells were washed with PBS, treated with RNase A (10 µg/mL), and stained with propidium iodide (PI, 50 µg/mL). The cell cycle of the stained cells was analyzed using flow cytometry (Attune NxT; Invitrogen, Waltham, MA, USA).

### 4.5. Cell Migration Assay

To quantify the extent of cell migration, 2D scratch assays and 3D Boyden chamber assays were conducted. For the scratch assay, a rectangular region within the VSMCs was scratched in each well using a cell scraper and treated with PDGF-BB or YHB. After 24 h, the cells were fixed and stained, and cell migration was determined based on the ratio of the wound closure area. For the Boyden chamber assay, the lower surface of the upper chamber was coated with 1% gelatin solution. Cells were seeded in the upper chamber at a concentration of 5 × 10^3^ cells/100 μL. YHB and inhibitors at predetermined concentrations were added to the upper chamber, and PDGF-BB was added to the lower chamber. The cells were then incubated at 37 °C for 8 h. After fixing the cells with methanol, the cells attached to the upper surface were removed using a cotton swab to observe only the cells that had migrated downward. The migrated cells were stained with crystal violet, and the number of migrated cells was measured in more than five areas per well to calculate the average value.

### 4.6. Immunocytochemistry

Cover glasses were placed in a 24-well plate, and cells were seeded at a density of 2 × 10^4^ cells/well in DMEM (containing 10% FBS) and cultured at 37 °C for 24 h. Subsequently, the cells were maintained under nutrient-deprived conditions with 0.1% FBS for 24 h. The cells were pretreated with YHB for 6 h, followed by a 24 h pre-treatment with 25 ng/mL PDGF-BB. For immunofluorescence labeling, the cells were rinsed with cold 1× PBS and fixed with 2% paraformaldehyde at RT for 15 min. After three washes with PBS, the cells were permeabilized or left untreated with 0.5% Triton-X-100 in PBS at RT for 15 min. Primary antibodies were incubated overnight at 4 °C in PBS containing 2% bovine serum albumin, followed by incubation with secondary antibodies at RT in the dark for 1 h. After washing with PBS, the cell nuclei were stained with 0.2 μg/mL DAPI (Sigma-Aldrich). Cover glasses were mounted on slides. Finally, the cells were visualized under a laser scanning confocal microscope (FluoView FV1000; Olympus, Tokyo, Japan).

### 4.7. Immunoblot Analysis

After seeding VSMCs at a density of 1 × 10^5^ cells/well in a 6-well plate, they were cultured for 24 h. The cells were maintained in a nutrient-deprived state using 0.1% FBS medium for an additional 24 h. YHB was then treated with various concentrations of YHB for 24 h. The medium was removed after 24 h, and the cells were washed twice with PBS. The cells were collected, lysed using RIPA buffer, and centrifuged at 14,000 rpm to separate proteins. The protein concentration was quantified using a Bradford protein assay kit (Bio-Rad, Hercules, CA, USA). Protein samples (40 μg) were subjected to 10% sodium dodecyl sulfate-polyacrylamide gel electrophoresis and then transferred to polyvinylidene difluoride membranes (Bio-Rad). The membrane was blocked with 5% skim milk at room temperature for 1 h and then incubated with primary antibodies overnight at 4 °C. The primary antibodies against the following were used: FAK (1:1000), p-FAK (Y397) (1:500), p-FAK (Y925) (1:500), Akt (1:4000), p-Akt (1:2000), ERK (1:3000), p-ERK (1:5000), P38 (1:2000), p-P38 (1:1000), mTOR (1:1000), p-mTOR (1:1000), C-Jun (1:1000), p-C-Jun (1:1000), FOXO3a (1:1000), PCNA (1:2000) (all from CST, Danvers, MA, USA), cyclin D1 (1:50), cyclin E (1:200), CDK4 (1:500), E2F1 (1:100) (all from Santa Cruz Biotechnology, Dallas, TX, USA), paxillin (1:2000; Millipore, Burlington, MA, USA), and β-actin (1:10,000; Sigma-Aldrich). Following primary antibody incubation, the membranes were washed three times with Tris-buffered saline-Tween 20 (TBS-T, 0.1% Tween 20) and then incubated with secondary antibodies for 1 h. After three washes, the protein bands were detected using an enhanced chemiluminescence reagent (ECL; BIONOTE, Animal Genetics Inc., Tallahassee, FL, USA).

### 4.8. Statistical Analysis

All quantified data were obtained at least three times and analyzed using GraphPad Prism 8.0 (GraphPad Software, La Jolla, CA, USA). Data are presented as mean ± SD, and statistical comparisons between two groups were performed using Student’s *t*-test. Statistical comparisons among multiple groups were performed using one-way analysis of variance (ANOVA), followed by the Bonferroni post hoc test when the F statistic was significant. A two-tailed *p* < 0.05 was considered statistically significant.

## Figures and Tables

**Figure 1 ijms-25-06899-f001:**
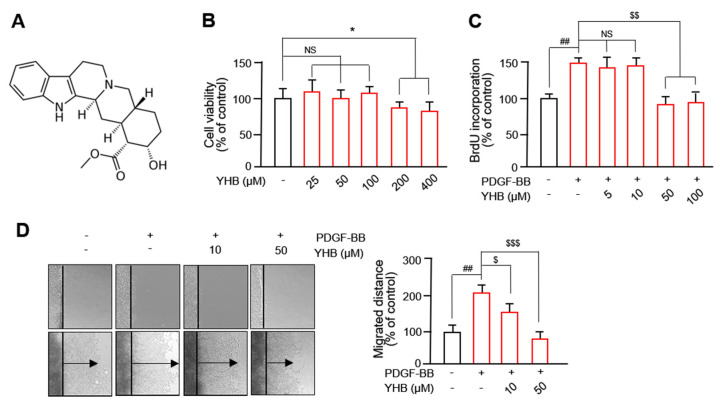
YHB inhibits PDGF-BB-induced VSMC proliferation and migration. (**A**) Chemical structure of YHB. (**B**) Cytotoxicity of YHB was evaluated using MTT assay. Serum-starved VSMCs were treated with YHB (25–400 μM) for 24 h. (**C**) Effect of YHB on VSMC proliferation was evaluated using BrdU incorporation assay. Serum-starved VSMCs were treated with YHB (5–100 μM) for 6 h and then incubated with PDGF-BB (25 ng/mL) for 24 h. (**D**) Effects of YHB on VSMC migration were measured using a wound healing assay. VSMCs were wounded (black lines) and treated with PDGF-BB with or without YHB. After 24 h, migrated cells were stained and photographed. The migrated distance was measured using ImageJ (https://imagej.net/ij/; accessed on 2 March 2024). Black arrow represents distance of the most migrated cells. All values are presented as mean ± standard deviation (SD) (n = 5). * *p* < 0.05 YHB versus control; ^##^
*p* < 0.01 PDGF-BB versus control; ^$^
*p* < 0.05, ^$$^
*p* < 0.01, and ^$$$^
*p* < 0.001 YHB versus PDGF-BB alone; NS, no significance.

**Figure 2 ijms-25-06899-f002:**
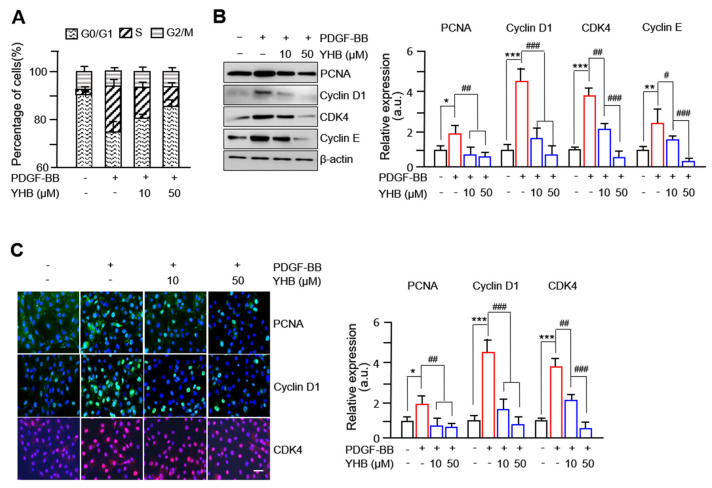
YHB inhibits PDGF-BB-mediated cell cycle progression and cell cycle regulatory proteins in VSMCs. (**A**) Effects of YHB on cell cycle progression. Serum-starved VSMCs were incubated with YHB (10–50 μM) for 2 h followed by 25 ng/mL of PDGF-BB treatment for 24 h. After DNA staining with propidium iodide, the cells were analyzed by flow cytometry. Each value shown was derived from counting >10,000 events and the numbers of cells (G0/G1, S, and G2/M phases) are expressed as % of 10,000 events (n = 5). (**B**) Inhibitory effect by YHB on PDGF-BB-stimulated expression of cell cycle regulatory proteins. Serum-starved VSMCs were incubated with YHB (10–50 μM) for 2 h, followed by PDGF-BB (25 ng/mL) treatment for 24 h. Band densities were normalized to those for β-actin. (**C**) Expression levels of cell cycle regulatory proteins (cyclin D1, green; CDK4, red) were evaluated using immunocytochemistry. Nuclei were stained with DAPI (blue). Scale bar, 200 μm. Magnification, 200×. All values are presented as mean ± SD (n = 5). * *p* < 0.05, ** *p* < 0.01, and *** *p* < 0.001 PDGF-BB versus control; ^#^
*p* < 0.05, ^##^
*p* < 0.01, and ^###^
*p* < 0.001 YHB versus PDGF-BB alone; NS, no significance.

**Figure 3 ijms-25-06899-f003:**
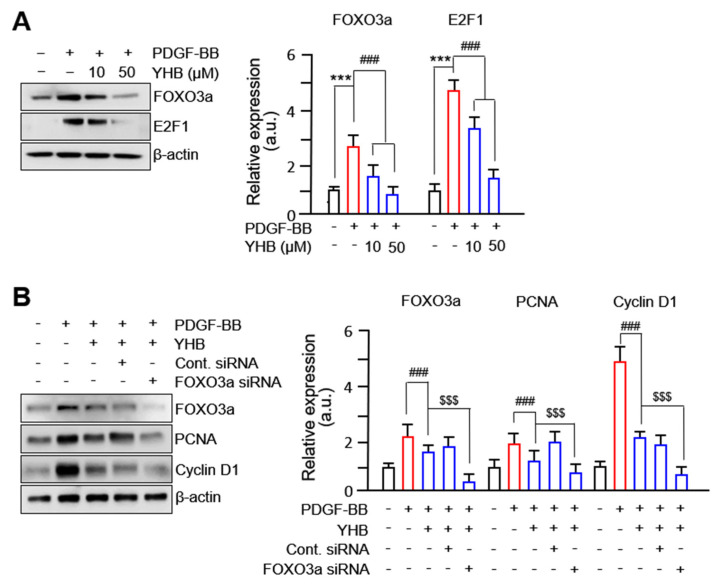
YHB suppresses PDGF-BB-induced VSMC proliferation via FOXO3a factor. (**A**) Effects of YHB on expression of FOXO3a and E2F-1. Serum-starved VSMCs were incubated with YHB (10–50 μM) for 2 h followed by 25 ng/mL of PDGF-BB treatment for 24 h. Band densities were normalized to those for β-actin. Representative images are shown from at least three independent experiments. (**B**) The levels of PCNA and cyclin D1 were evaluated in FOXO3a-silenced VSMCs. After 24 h of siRNA transfection, VSMCs were incubated with YHB (10–50 μM) for 2 h followed by 25 ng/mL of PDGF-BB treatment for 24 h. Band densities were normalized to those for β-actin. Representative images are shown from at least three independent experiments. All values are presented as mean ± SD (n = 5). *** *p* < 0.001 PDGF-BB versus control; ^###^
*p* < 0.001 YHB versus PDGF-BB alone; ^$$$^
*p* < 0.001 FOXO3a siRNA-transfected versus non-transfected; NS, no significance.

**Figure 4 ijms-25-06899-f004:**
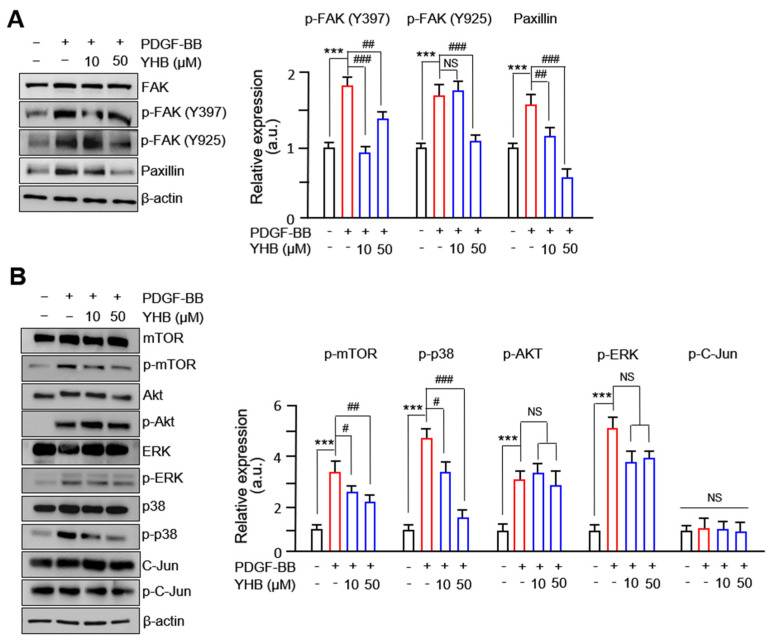
YHB attenuates FAK-related and mTOR/p38 signaling pathways upon PDGF-BB treatment. (**A**) Effects of YHB on expression of FAK-related proteins in VSMCs. Serum-starved VSMCs were incubated with YHB (10–50 μM) for 2 h followed by 25 ng/mL of PDGF-BB treatment for 24 h. (**B**) Effects of YHB on expression of MAPK. Serum-starved VSMCs were incubated with YHB (10–50 μM) for 2 h followed by 25 ng/mL of PDGF-BB treatment for 30 min. Band densities phosphorylated proteins were normalized to those of total protein expression. All values are presented as mean ± SD (n = 5). *** *p* < 0.001 PDGF-BB versus control; ^#^
*p* < 0.05, ^##^
*p* < 0.01, and ^###^
*p* < 0.001 YHB versus PDGF-BB alone; NS, no significance.

**Figure 5 ijms-25-06899-f005:**
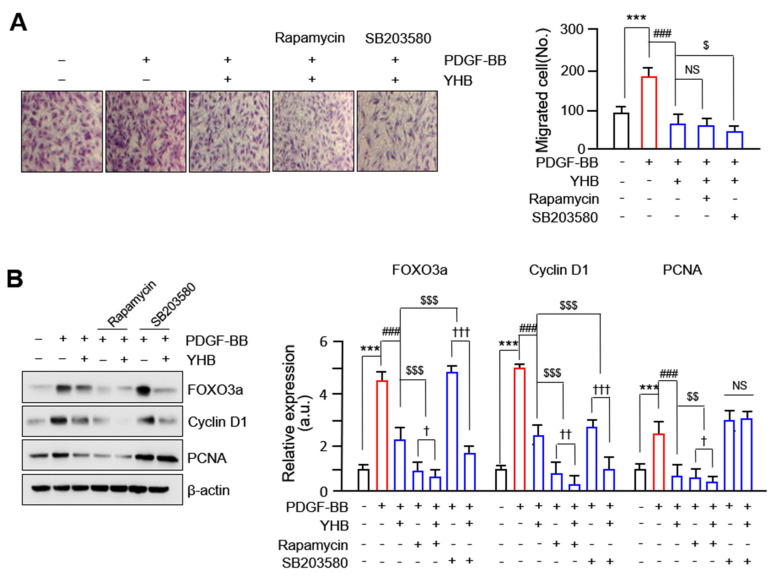
Synergistic effect of YHB with mTOR/p38 signaling inhibition in PDGF-BB-stimulated VSMCs. (**A**) Boyden chamber assay showing number of migrated cells. Representative images are shown from at least five independent experiments, taken 8 h after seeding. Scale bar, 200 μm. Magnification, 200×. (**B**) Effects of YHB with or without MAPK inhibitors on cell proliferation regulatory proteins in PDGF-BB-stimulated VSMCs. Band densities were normalized to those for β-actin. All values are presented as mean ± SD (n = 5). *** *p* < 0.001 PDGF-BB versus control; ^###^
*p* < 0.001 YHB versus PDGF-BB alone; ^$^
*p* < 0.05, ^$$^
*p* < 0.01, and ^$$$^
*p* < 0.001 YHB alone versus each inhibitor; ^†^
*p* < 0.05, ^††^
*p* < 0.01, and ^†††^
*p* < 0.001 YHB alone versus co-treatment of YHB with each inhibitor; NS, no significance.

**Figure 6 ijms-25-06899-f006:**
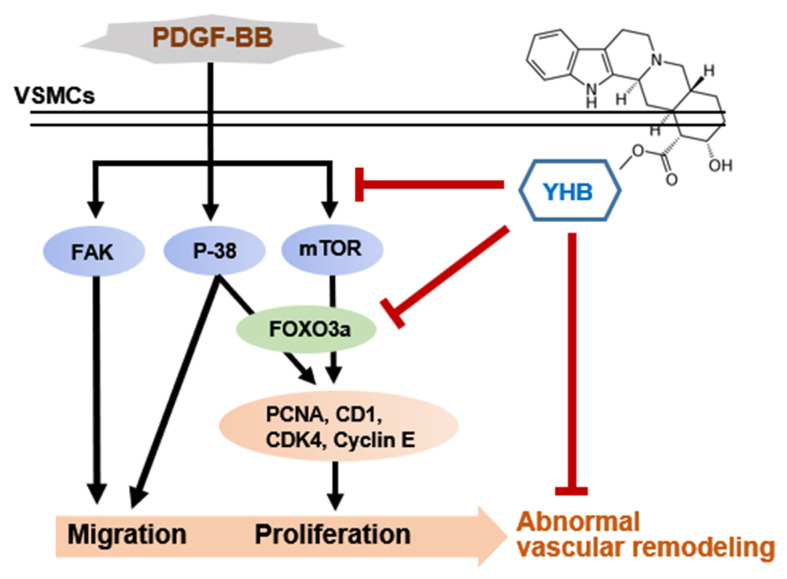
Proposed scheme for inhibitory effects of YHB on PDGF-BB-induced proliferation and migration of VSMCs.

## Data Availability

The data presented in this study are available upon request from the corresponding author. The data are not publicly available due to privacy concerns.

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
