# Peer review of "Yohimbine Inhibits PDGF-Induced Vascular Smooth Muscle Cell Proliferation and Migration via FOXO3a Factor"

_ijms, 2024, doi:10.3390/ijms25136899_

Round 1
Reviewer 1 Report
Comments and Suggestions for Authors
The manuscript by Lim et al describes a novel effect of Yohimbine (YHB) on vascular smooth muscle cell (VSMC) biology and signalling by supressing PDGF-BB induced activation of mTOR signalling via FOXO3a. The data might indicate that the substance has a potential to be used as a prevention in the therapy of diseases linked to VSMC activity and proliferation. However, some issues have to be clarified:
1. the authors need to provide information how many rats or VSMC cell lines were used or if experiments were performed repeatedly in one cell line.
2. please explain the term "PI" in line 360. It is assumably propidiumiodide.
3. all experiments were performed in PDGF stimulated VSMC. It would be of interest to see if YHB had any significant effect on base line protein activity or expression. Such data provide important information about the suggested use as a preventive or curative drug in the future.
4. in figure 3B (Western-blot) the effect of control FOXO3a siRNA on FOXO3a itself as well as on cyclin D1 is nearly as strong as that of FOXO3a siRNA. Can the blot be improved?
5. in the graphic abstract it is suggested that three different signalling pathways are blocked by YHB (p38, mTOR,. However, referring to other publications on signalling cascades, FAK is most often activated before p38 MAPK and before mTOR. Furthermore, FOXO3a has also been reported to downregulate mTOR via Rictor, thus, the suggested interference of YHB with mTOR signalling and the other pro-proliferative proteins should have been assessed in the presence of either rapamycin or the FOXO3a siRNA as well. Do the authors have any such data to be added?
6. in the discussion the authors suggest to use YHB for prevention and therapy of VSMC associated diseases. The data only provides the basis for a preventive effect, but not for a therapeutic one. To suggest such an effect the same experiments should have been performed with either YHB applied at the same time as PDGF or YHB given 30-60 minutes after PDGF. in the diseased state the inflammation is already active. This should be at least discussed.
Comments on the Quality of English LanguageThe English is fine, but needs some minor editing.
Author Response
We thank the reviewer for the positive comments and detailed review: “The manuscript by Lim et al describes a novel effect of Yohimbine (YHB) on vascular smooth muscle cell (VSMC) biology and signalling by supressing PDGF-BB induced activation of mTOR signalling via FOXO3a. The data might indicate that the substance has a potential to be used as a prevention in the therapy of diseases linked to VSMC activity and proliferation. However, some issues have to be clarified.”
Our responses to the reviewer’s suggestions/comments are detailed below:
- the authors need to provide information how many rats or VSMC cell lines were used or if experiments were performed repeatedly in one cell line.
VSMCs, which was one of the cell lines used in the study, were primarily isolated from rat aorta, as described in the Materials and Methods section.
- please explain the term "PI" in line 360. It is assumably propidiumiodide.
We have made the necessary revision, as suggested.
- all experiments were performed in PDGF stimulated VSMC. It would be of interest to see if YHB had any significant effect on base line protein activity or expression. Such data provide important information about the suggested use as a preventive or curative drug in the future.
We agree with the reviewer's opinion that if YHB has a significant effect on baseline protein activity or expression, it can be used as a preventive drug. However, we did not observe any significant changes in the levels of proteins related to proliferation or migration after treatment with YHB alone. As the migration and proliferation of VSMCs are induced not only by PDGF but also by various stimuli, our results demonstrate that the inhibitory effect of YHB is likely specific to PDGF-stimulated VSMCs.
- in figure 3B (Western-blot) the effect of control FOXO3a siRNA on FOXO3a itself as well as on cyclin D1 is nearly as strong as that of FOXO3a siRNA. Can the blot be improved?
The decrease in cyclin D1 in Figure 3B was not due to a non-specific effect of FOXO3a siRNA, but due to FOXO3a silencing.
- in the graphic abstract it is suggested that three different signalling pathways are blocked by YHB (p38, mTOR,. However, referring to other publications on signalling cascades, FAK is most often activated before p38 MAPK and before mTOR. Furthermore, FOXO3a has also been reported to downregulate mTOR via Rictor, thus, the suggested interference of YHB with mTOR signalling and the other pro-proliferative proteins should have been assessed in the presence of either rapamycin or the FOXO3a siRNA as well. Do the authors have any such data to be added?
Our results confirmed that FAK and p38 inhibition by YHB occurred independently, and further research is needed to determine the interrelationship between them. Unfortunately, in the present study, we were unable to conduct experiments on the hierarchical relationship between FOXO3a and mTOR or on proteins that promote proliferation. However, we are conducting other experiments to confirm the effects of YHB in animal models of atherosclerosis. We plan to clarify the signaling system in more detail in our future studies.
- in the discussion the authors suggest to use YHB for prevention and therapy of VSMC associated diseases. The data only provides the basis for a preventive effect, but not for a therapeutic one. To suggest such an effect the same experiments should have been performed with either YHB applied at the same time as PDGF or YHB given 30-60 minutes after PDGF. in the diseased state the inflammation is already active. This should be at least discussed.
As suggested by the reviewer, we could not determine the therapeutic effects of YHB in vitro. We plan to demonstrate the therapeutic effects of YHA in future animal studies. Therefore, the corresponding text under the Discussion section has been modified as follows: “Further experiments on animal models are warranted to determine the therapeutic effect of YHB in cases where atherosclerosis is induced. Moreover, future animal model studies should also focus on cross validating the signaling system presented in this study.”
Reviewer 2 Report
Comments and Suggestions for Authors
The authors conducted a serious study to indicate the the inhibitory effects of YHB on the proliferation and migration of PDGF-BB-stimulated VSMCs by regulating the transcription factor FOXO3a and the mTOR/p38/FAK signaling pathway. The findings may provide a potential therapeutic candidate for preventing and treating cardiovascular diseases such as atherosclerosis and vascular restenosis. The experiments were well designed. However, there are still several points needs to be addressed before the acceptance for publication.
Comments:
1. Figure 2C, for the immunocytochemistry results, the cells were stained green, blue, and red, but it was not denoted in the figure legend. Please supplement the related description.
2. β-actin was used as the reference gene for all the immunoblot analysis in this study. Is it an ideal reference gene for the present study? And does it consistently express during VSMCs proliferation and migration processes?
For the immunoblot analysis displayed in this study, the protein level of β-actin was high in Figure 3, but much lower in Figure 4B. And in Figure 4B, its protein looks higher when the cells were treated with PDGF-BB. This situation continued when combined with 10 μM YHB.
Author Response
We thank the reviewer for the positive comments and detailed review: “The authors conducted a serious study to indicate the the inhibitory effects of YHB on the proliferation and migration of PDGF-BB-stimulated VSMCs by regulating the transcription factor FOXO3a and the mTOR/p38/FAK signaling pathway. The findings may provide a potential therapeutic candidate for preventing and treating cardiovascular diseases such as atherosclerosis and vascular restenosis. The experiments were well designed. However, there are still several points needs to be addressed before the acceptance for publication.”
Our responses to the reviewer’s suggestions/comments are detailed below:
- Figure 2C, for immunocytochemistry, the cells were stained green, blue, and red, but not shown in the figure legend. Please supplement with the related descriptions.
As the reviewer suggested, we have edited the figure legend (Figure 2C) and added a description for different colors depicted in the immunocytochemical staining results in the figure.
- β-actin was used as the reference gene for all the immunoblot analysis in this study. Is it an ideal reference gene for the present study? And does it consistently express during VSMCs proliferation and migration processes?
For the immunoblot analysis displayed in this study, the protein level of β-actin was high in Figure 3, but much lower in Figure 4B. And in Figure 4B, its protein looks higher when the cells were treated with PDGF-BB. This situation continued when combined with 10 μM YHB.
In immunoblots, b-actin, tubulin, and Gapdh are mainly used as reference genes, and actin is generally used as a reference gene in VSMC migration and proliferation studies. In addition, we observed that the expression levels of b-actin remained consistent under our experimental conditions, regardless of PDGF treatment. Typically, the size of the actin band varies in each experiment owing to differences in the exposure time during immunoblot development. Although the level of b-actin expression did not change significantly upon PDGF-BB treatment, the image of b-actin in Fig. 4B has been replaced.